

# Tibyan corpus: balanced and comprehensive error coverage corpus using ChatGPT for Arabic grammatical error correction

Ahlam Alrehili[1,2] and Areej Alhothali[1]

[1] Department of Computer Sciences, Faculty of Computing and Information Technology, King Abdul Aziz University, Jeddah, Saudi Arabia
[2] Department of Computer Science, College of Computing and Informatics, Saudi Electronic University, Riyadh, Saudi Arabia

## ABSTRACT

Natural language processing (NLP) augments text data to overcome sample size constraints. Scarce and low-quality data present particular challenges when learning from these domains. Increasing the sample size is a natural and widely used strategy for alleviating these challenges. Moreover, data-augmentation techniques are commonly used in languages with rich data resources to address problems such as exposure bias. In this study, we chose Arabic to increase the sample size and correct grammatical errors. Arabic is considered one of the languages with limited resources for grammatical error correction (GEC) despite being one of the most popular among Arabs and non-Arabs because of its close connection to Islam. Therefore, this study aims to develop an Arabic corpus called "Tibyan" for grammatical error correction using ChatGPT. ChatGPT is used as a data augmenter tool based on a pair of Arabic sentences containing grammatical errors matched with a sentence free of errors extracted from Arabic books, called guide sentences. Multiple steps were involved in establishing our corpus, including collecting and pre-processing a pair of Arabic texts from various sources, such as books and open-access corpora. We then used ChatGPT to generate a parallel corpus based on the text collected previously, as a guide for generating sentences with multiple types of errors. By engaging linguistic experts to review and validate the automatically generated sentences, we ensured they were correct and error-free. The corpus was validated and refined iteratively based on feedback provided by linguistic experts to improve its accuracy. Finally, we used the Arabic Error Type Annotation tool (ARETA) to analyze the types of errors in the Tibyan corpus. Our corpus contained 49% of errors, including seven types: orthography, morphology, syntax, semantics, punctuation, merge, and split. The Tibyan corpus contains approximately 600 K tokens.

# INTRODUCTION

The Arabic language has a great deal of influence worldwide. It is an ancient language with deep roots in human history. The Holy Qur'an's language is Arabic, which has a special

Corresponding author
Ahlam Alrehili,
ahm.alrehili@gmail.com

religious status among Muslims worldwide. Many of the most significant literary and philosophical works in human history have been written in Arabic, making it a sophisticated literary and poetic language (*Bakalla, 2023*). In addition to being an important scientific and intellectual language, Arabic has contributed greatly to the transfer of knowledge and culture to Europe and other countries as it was the language of scholars and philosophers during the Middle Ages (*Chejne, 1968*).

One of the most important and famous features of Arabic is that it consists of three main versions: classical Arabic, Modern Standard Arabic (MSA), and regional dialects (*Ferguson, 1959*). Classical Arabic was used in the Holy Quran and ancient literary texts between the 7th and 9th centuries (*Holes, 2004*). Non-native Arabic speakers may find it difficult to learn classical Arabic or Quranic Arabic because of the special symbols (Tanween) that indicate proper pronunciation. MSA is the official language used primarily in newspapers, television broadcasts, and films. As it is not commonly spoken as a first language, it is a language without native speakers. There are currently 274 million speakers worldwide (https://www.statista.com/statistics/266808/the-most-spoken-languages-worldwide/). MSA is a formal language that is not used in daily life. Arabic is a vast language with a variety of dialects, and all Arabic speakers learn a local dialect, such as Mesopotamian Arabic and Egyptian Arabic. Meanwhile, dialectal Arabic is used by Arabs as a daily language of conversation. Although Arabic dialects are fundamentally related, they cannot be understood by one another because Arab countries speak different dialects (*Holes, 2004*).

Because MSA is rarely used in daily life, it is sometimes mixed with local dialects. Moreover, because of the rich and intricate nature of Arabic, ambiguity can lead to incomprehensible and inaccurate text. In addition, Arabic grammar presents several semantic, syntactic, and morphological challenges owing to its flexible word order, diacritic, and agglutination properties. Furthermore, considerable deficiencies at the Arabic morphological level have hampered extensive research in this area. At higher research levels, semantics and syntax did not significantly advance. Therefore, GEC is becoming increasingly important for native and non-native speakers.

GEC automatically detects and corrects grammatical errors in a text (*Bryant et al., 2023*). Recent approaches to grammatical error correction, such as the seq2seq model, require large, high-quality parallel datasets. However, many languages do not contain such data, making it difficult to train these models. Other languages contain only a limited number of examples, making it difficult to build models that can correct all types of linguistic errors. Moreover, the creation of such datasets can be time-consuming and expensive. Therefore, most researchers use data augmentation techniques to increase the size of GEC parallel data. Data augmentation techniques generate more diverse training examples, which enhances the model's ability to generalize to unknown errors. Various error types and contexts can be introduced using data augmentation to balance the dataset. Consequently, GEC systems have become more robust and accurate.

The Arabic language has limited resources. Only two parallel corpora are available for GEC research: QALB-14 (*Mohit et al., 2014*) and QALB-15 (*Rozovskaya et al., 2015*). The QALB-14 and QALB-15 are part of the Qatar Arabic Language Bank (QALB) project.

QALB aims to create a large corpus of Arabic texts that have been manually corrected, such as user comments on news sites, essays written by native and non-native speakers, and machine translation text. A specialized annotation interface was developed for this project, along with comprehensive annotation guidelines (*Jeblee et al., 2014*; *Obeid et al., 2013*). A total of 20,430 and 1,542 samples were available from the two training corpora, (QALB-14) and (QALB-15). Despite the researchers' complete reliance on these data, they have some shortcomings, including inadequate coverage of Arabic language defects, inconsistent punctuation correction, and small size compared with other datasets in other languages.

This study aims to contribute to the development of an Arabic corpus for grammatical error correction by employing ChatGPT to generate paired sentences based on common errors found in Arabic books. First, we collected a diverse range of pair Arabic sentences; one containing common grammatical errors made by native speakers and other corrected versions of the sentence. The sentences collected from the three Arabic books were short, ranging from one to seven words. These sentences were extracted from three Arabic books namely "A Dictionary of Common Grammatical, morphological, and Linguistic Errors" (https://archive.org/details/20210306_20210306_1934/mode/2up), "Common linguistic errors in cultural circles" (*Alrehili & Alhothali, 2024*), "Common linguistic errors" (https://www.alukah.net/books/files/book_5755/bookfile/akhtaa.pdf). Moreover, we used the A7'ta corpus (*Madi & Al-Khalifa, 2019*) which is composed of 466 short sentence pairs taken from a book called Linguistic Error Detector (Saudi Press). Second, we instructed the ChatGPT model to generate full sentence pairs using our collected short sentence pairs, one containing the error and the other free from errors. Additionally, the corrected versions of the corpus were annotated and all grammatical errors were corrected by experts, creating a valuable resource for training and evaluating the performance of the Arabic GEC. Finally, we analyzed the types of errors generated in our corpus using the ARETA tool (*Belkebir & Habash, 2021*). We make our corpus publicly available. The contributions of this study are as follows.

- Collect and organize short Arabic sentences, including common grammatical errors, from various Arabic books as a guide.
- Using ChatGPT as a data augmenter, a full, long, and error-free Arabic corpus can be generated from the guiding sentences, resulting in an error-prone Arabic corpus.
- Assuring that annotated errors are accurate and relevant by engaging linguistic experts to review and validate them manually.
- The corpus was validated and refined iteratively based on the feedback provided by linguistic experts. Understanding the distribution and characteristics of errors in different contexts by analyzing the linguistic properties of the corpus.

The remainder of this artcile is organized as follows: first, we discuss the available Arabic corpora and studies that used ChatGPT as a data aggregator. Next, we describe the methodology used to build the GEC corpus. Then, we describe our experimental setup. Subsequently, we analyze the type and percentage of errors in our corpus. Next, we

describe the application implications and limitations of our corpus. Finally, we summarize our contributions and outline future directions for Arabic GEC research.[1]

[1] Portions of this text were previously published as part of a preprint (*Alrehili & Alhothali, 2024*).

# RELATED WORK

This section discusses the Arabic corpus available for GEC and recent research using ChatGPT as a data augmentation for GEC.

## Arabic corpus

Five Arabic GEC datasets are publicly available for grammatical error correction. The first two are derived from the shared QALB-14 (*Mohit et al., 2014*) and QALB-15 (*Rozovskaya et al., 2015*) tasks. In addition to these, there are the A7'ta corpus (*Madi & Al-Khalifa, 2019*), the ZAEBUC dataset (*Habash & Palfreyman, 2022*) and Lang-8 corpus (*Mizumoto et al., 2011*). None of them were manually annotated for a specific type of error. An overview of the dataset statistics is presented in Table 1.

The QALB corpus is one of the components of the QALB project and was created as part of it. In the QALB project, large manual correction corpora for a variety of Arabic texts were developed, including texts written by native and non-native authors and machine translation outputs.

The QALB-14 (*Mohit et al., 2014*) is a compilation of MSA comments written by native speakers on the Al Jazeera News website. Both native and non-native Arabic speakers were addressed in QALB-15. Learners of Arabic as a second language (L2) contributed texts to the QALB-15 (*Rozovskaya et al., 2015*), extracted from two learner corpora: the Arabic Learner Corpus (ALC) (*Alfaifi & Atwell, 2012*) and the Arabic Learners Written Corpus (ALWC) (*Farwaneh & Tamimi, 2012*). The annotation process was divided into three phases: automatic preprocessing, automatic spelling corrections, and manual annotation by humans (annotators). They used morphological analysis (*Habash & Rambow, 2005*) and the disambiguation system MADA (version 3.2) (*Habash, Rambow & Roth, 2009*) to automate spelling corrections. Annotators were required to correct spelling, punctuation, word choice, morphology, syntax, and dialect errors. There were 21,396 sentences in the QALB-14 Corpus and 1,533 sentences in the QALB-15 corpus, divided into training, development, and test sentences.

The QALB corpus contains valuable Arabic data but not all types of errors, such as lengthening short vowels, Nun dan Tanwin confusion, and shortening long vowels (*Belkebir & Habash, 2021*). The datasets contained inconsistent manual annotations of punctuation corrections; for example, there was a space between the full stop and the word.

A7'ta (*Madi & Al-Khalifa, 2019*) is a parallel monolingual corpus that presents Arabic texts in parallel. A total of 470 erroneous sentences and 470 correct sentences were found. Sentences were collected manually from a book called the Linguistic Error Detector (Saudi Press), which was designed to guide writers and readers in correct Arabic grammar usage. In this corpus, there are only 3,532 tokens, the majority of which are incomplete sentences, which cannot be used alone for deep learning.

ZAEBUC (*Habash & Palfreyman, 2022*) is a bilingual corpus annotated in Arabic and English by first-year university students at Zayed University. Designed to represent

**Table 1 Available Arabic parallel corpus.**

| Corpus | Split | Line | Words | Level | Domain |
|--------|-------|------|-------|-------|--------|
| QALB-14 | Train | 19.4 K | 1 M | L1 | Comments |
| | Dev | 1 K | 54 K | L1 | Comments |
| | Test | 948 | 51 K | L1 | Comments |
| QALB-15 | Train | 310 | 43.3K | L2 | Essays |
| | Dev | 154 | 24.7 K | L2 | Essays |
| | Test-L1 | 158 | 22.8 K | L2 | Essays |
| | Test-L2 | 940 | 48.5 K | L2 | Comments |
| ZAEBUC | No spilt | 214 | 33,376 | L1 | Essays |
| A7'ta | No spilt | 466 | – | L1 | Sentences |
| lang-8 | No spilt | 737 | – | L2 | Comments |

bilingual writers, one writing in their native language and one writing in their second language, the corpus contained short essay bilingual corpora matched to writers. The corpus creation process involved four steps. The first step was to obtain approval from ZU's IRB board and then contact the faculty teaching the targeted courses. Written consent was obtained from all participating students. In parallel to the second step, manual text correction and CEFR annotation were performed independently. Morphological annotation followed text correction depending on the results. Finally, the semi-automatic annotations were manually corrected. There were 214 sentences in total, which was a relatively small corpus.

The Lang-8 corpus (*Mizumoto et al., 2011*) ranks as one of the largest corpora for training grammatical error correction systems based on machine translation. Furthermore, it contains nearly 80 languages of learners and corrected sentences based on Lang-8's (https://lang-8.com/) revision logs. There are approximately 737 sentence pairs in Arabic, which is one of the top 20 languages. However, the Lang-8 corpus is not suitable for evaluation because annotators do more than correct a learner's sentence; it also provides feedback. Learners can benefit from these comments. However, the comments were merely noise in an evaluation dataset. Moreover, it is possible to find Arabic texts mixed with the learners' language.

## ChatGPT for data augmentation

ChatGPT has recently demonstrated effective GEC performance using zero-shot and few-shot prompts (*Wu et al., 2023*; *Fang et al., 2023*; *Loem et al., 2023*). ChatGPT is used in GEC in various ways. For example, *Zhang et al. (2023)* evaluated the effectiveness of ChatGPT as a corrector for GEC by using a prompt-based approach. In-structured ChatGPT to correct sentences for grammatical errors. In this study, perturbations unrelated to errors were introduced into ChatGPT to evaluate the context robustness.

Moreover, ChatGPT used as a data augmenter, such as in *Fan et al. (2023)*, introduced GrammarGPT, an open-source Large Language Model (LLM) that is designed to correct native Chinese grammar errors. *Fan et al. (2023)* studied ChatGPT-generated and

human-annotated datasets in conjunction with an error-invariant augmentation method to achieve better accuracy when correcting native Chinese grammatical errors. They guided ChatGPT in generating ungrammatical sentences by providing clues and manually correcting sentences collected from websites without clues. The model was enhanced to correct native Chinese grammatical errors using an error-invariant augmentation method. A hybrid dataset of ChatGPT-generated and human-annotated data was used to fine-tune open-source LLMs with instruction tuning. Native Chinese grammatical error correction using open-source LLMs was demonstrated using this approach. In addition, it can be used to introduce natural language explanations for correction reasons. *Kaneko & Okazaki (2023)* introduced a method called Controlled Generation with Prompt Insertion (PI) that allows LLMs to explain the reasons for corrections in natural language in the context of GEC. The GEC explanations were improved using Chat-GPT. LLMs are used to explain the correction reasons in natural language using ChatGPT in a technique called controlled generation with PI. The LLMs produced better correction reasons by inserting edit prompts during generation and explicitly engaging them in providing explanations for all edits. According to the study, PI led to enhanced performance when describing the correction reasons for all correction points compared to using the original prompts for generation.

The only Arabic study that has used ChatGPT to augment data is that of *Kwon et al. (2023)*, who used ChatGPT to inject grammatical errors into Arabic text. They created a parallel dataset using ChatGPT by selecting and corrupting 10,000 correct sentences from an original training set. Therefore, note that our approach to increasing the amount of data using ChatGPT is unique.

To the best of our knowledge, this is the first study to augment data by extracting sentence fragments from books (guide sentences) and instructing ChatGPT to generate two sentences using guide sentences, one correct and one with errors. According to an extensive review, there is a lack of research utilizing similar techniques for data augmentation. A novel avenue for expanding datasets was created by leveraging ChatGPT, which holds considerable promise across several fields. In addition to enriching the available data, this innovative method illustrates the versatility and adaptability of ChatGPT. Exploring and validating this approach can significantly advance this field and open doors for new possibilities and insights.

## APPROACH

Figure 1 shows the proposed approach. We began by collecting pairs of sentences from Arabic books, one of which was correct and the other contained grammatical errors. This is called a guide sentence. There is also an Arabic corpus called a7'ta that contains sentences extracted from Arabic books. Guide sentences are usually short with limited tokens and incomplete sentences. ChatGPT was then instructed to construct two useful sentences based on the guide sentences: one with correct guide sentences and the other with grammatical errors based on incorrect guide sentences. In addition, the data were reviewed by a human annotator to ensure that they were accurate and did not contain grammatical errors.

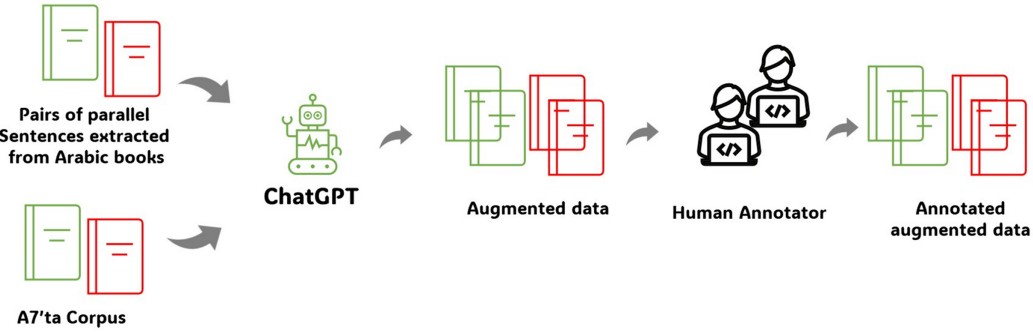

**Figure 1** **Process of creating the Tibyan corpus.**

## Data collection

Owing to the lack of parallel Arabic corpora, we initially collected pairs of correct and incorrect sentences. Various sources including books and available corpus were used during this phase. The following are the descriptions of the three Arabic books used:

- **A Dictionary of Common Grammatical, morphological, and Linguistic Errors**: Several common linguistic errors are highlighted in this dictionary book to alert Arabic language students. Four main types of errors are discussed in "A Dictionary of Common Grammatical, morphological, and Linguistic Errors": Syntactic errors, errors in transitive verbs with prepositions, errors in grammar, morphology, and sentence structure, and errors in correctness and semantics.
- **Common linguistic errors in cultural circles**: It contains six types of errors, which include errors in syntax, nouns, verbs, linguistic structures, masculine and feminine, and phonetics.
- **Common linguistic errors**: This dataset contains approximately 83 sentence pairs with the most common linguistic errors in Arabic.

Table 2 lists the number of sentences and types of errors included in each book. The total number of sentences was 3,166. In addition, there is an available corpus called A7'ta (*Madi & Al-Khalifa, 2019*), which contains 466 sentence pairs extracted from a linguistic error detector (Saudi Press). It contains eight types of errors: syntactic, morphological, semantic, linguistic, stylistic, spelling, punctuation, and the use of informal and borrowed words (*Madi & Al-Khalifa, 2019*).

## Data pre-processing

Preprocessing was performed once valuable linguistic sources were gathered. We manually extracted sentences from these sources from the hand sides of the books. As shown in Fig. 2, the books contained correct and incorrect sentences along with explanations and clarifications. A separate file was created for each correct and incorrect sentence pair. One file contained the correct sentences, and the other contained incorrect sentences. Several obstacles were encountered, including the existence of correct sentences without incorrect sentences. In this case, we repeated the correct sentence in the files of correct sentences and

**Table 2 Arabic books used in the data collection phase.**

| Book | # Sentence | Errors type |
|---|---|---|
| A dictionary of common grammatical, morphological, and linguistic errors | 2,241 | Syntactic errors |
| | | Verbs with prepositions errors |
| | | Grammatical errors |
| | | Morphology |
| | | Sentence structure |
| | | Semantics errors |
| Common linguistic errors in cultural circles | 842 | Syntax errors |
| | | Nouns errors |
| | | Verbs errors |
| | | Linguistic structures errors |
| | | Masculine and feminine errors |
| | | Phonetic errors |
| Common linguistic errors | 83 | Syntactic errors |
| | | Grammatical errors |
| | | Morphological errors |
| | | Punctuation errors |

incorrect sentences to increase the sample size and avoid ignoring any errors. In some cases, there was more than one correct sentence equivalent to one incorrect sentence; therefore, all correct sentences were placed in separate lines, and incorrect sentences were repeated for each correct sentence.

A7'ta corpus has 300 folders. Each book's eight main categories were further divided into eight categories. For each subcategory within the main category, there are many subfolders within the folder for each error type. Each error-type folder contains two files: one for correctly written sentences (correctness) and another for erroneous sentences (error). Sentence pairs were manually extracted from all folders. We then saved them in two separate files: one containing errors, and the other containing correct sentences.

## Data augmentation

The data collected in the previous stage consisted of sentences of one to eighteen words, as shown in Fig. 3. The average word length was four words. At least one word differed between the sentence pairs. Moreover, it can be expressed as part of a sentence or as an incomplete sentence. These data are not valuable for many modern approaches such as seq2seq (*Sutskever, Vinyals & Le, 2014*) and seq2edit (*Stahlberg & Kumar, 2020*), which require large amounts of data. Therefore, we used ChatGPT to convert parts of the sentences into full sentences. ChatGPT is a machine learning and artificial neural network-based artificial intelligence language model. ChatGPT supports advanced natural-language understanding and generation, making it useful for a wide range of applications. ChatGPT was used for creative text synthesis, writing assistance, content generation, translation, and natural interactions with users in chatbots. Several areas of artificial intelligence can benefit

أَبَدًا، وَقَطُّ:

Correct

قُلْ: لَنْ أَكْذِبَ أَبَدًا.
وَقُلْ: لَمْ أَكْذِبْ قَطُّ.

Incorrect

لَا تَقُلْ: لَنْ أَكْذِبَ قَطُّ.
وَلَا تَقُلْ: لَمْ أَكْذِبْ أَبَدًا.

Explanation

التَّحْلِيلُ: يُسْتَخْدَمُ الظَّرْفُ «قَطُّ» لِلنَّفْيِ فِي الْمَاضِي، وَفِي أَحْيَانٍ نَادِرَةٍ يُسْتَخْدَمُ فِي غَيْرِ النَّفْيِ. كَمَا يُسْتَخْدَمُ الظَّرْفُ «أَبَدًا» لِلنَّفْيِ فِي الْمُسْتَقْبَلِ، كَمَا يُسْتَخْدَمُ قَلِيلًا فِي غَيْرِ النَّفْيِ، وَلَا يَجُوزُ أَنْ يُسْتَخْدَمَ أَحَدُهُمَا فِي مَوْضِعِ الْآخَرِ إِلَّا إِذَا أَمْكَنَ تَأْوِيلُ الْمَاضِي مُسْتَقْبَلًا، وَفِي هَذِهِ الْحَالَةِ يُمْكِنُ أَنْ نَسْتَخْدِمَ «أَبَدًا» مَعَ نَفْيِ الْمَاضِي. وَقَدْ وَرَدَ هَذَا فِي عَدِيدٍ مِنْ كُتُبِ اللُّغَةِ، وَمِنْ بَيْنِهَا «مُعْجَمُ قَوَاعِدِ اللُّغَةِ» لِعَبْدِ الْغَنِيِّ الدَّقْرِ إِذْ يَقُولُ فِي مَعْنَى «أَبَدًا»: «وَلَا يَدْخُلُ عَلَى الْمَاضِي إِلَّا إِذَا كَانَ الْمَاضِي مُمْتَدًّا إِلَى الْمُسْتَقْبَلِ نَحْوَ قَوْلِهِ (تَعَالَى): ﴿ وَبَدَا بَيْنَنَا وَبَيْنَكُمُ الْعَدَاوَةُ وَالْبَغْضَاءُ أَبَدًا حَتَّىٰ تُؤْمِنُوا بِاللَّهِ ﴾ (الْمُمْتَحَنَةُ: 4).

**Figure 2** Sample of data exist in books.

Correct

ويطمح
بل يطمحون
متمنياً له ولمرافقيه
ويساعد هو وزوجته أم كلثوم
الجوهرة بنت
فاعلا أساسيا
في البحرين، ولبنان، وسورية واليمن

Incorrect

بل ويطمح
بل ويطمحون
متمنيا له ومرافقيه
ويساعد وزوجته أم كلثوم
الجوهرة بن
كفاعل أساس
في البحرين، لبنان، ثم سورية واليمن

**Figure 3** Sample of data after data collection phase.

from the ChatGPT technology, which represents a qualitative leap in natural language understanding.

In this study, we employ ChatGPT to augment parallel data for grammatical error correction. By providing ChatGPT with the correct partial sentences obtained during data collection, we instructed it to construct a complete and useful sentence. We then instructed ChatGPT to replace the correct sentence fragment with an incorrect sentence fragment and generate grammatical errors, resulting in parallel data. Data were stored in two separate files. The first file contains a generated sentence containing the correct fragment, whereas the second file contains the same sentence but with the correct fragment replaced with an incorrect fragment, and contains grammatical errors. Using this innovative

approach, we can enhance the ability of linguistic models to understand and generate texts better and more accurately, which enhances the quality and diversity of data. The flowchart in Fig. 4 shows a parallel corpus created by using ChatGPT as a data augmentation tool.

Using ChatGPT, we improved data augmentation by combining manual and automated approaches. The research employed two OpenAI models in this study: GPT-3.5 for automated sentence generation and GPT-4 for a manual approach *via* the ChatGPT interface.

In the manual approach, we manually provided ChatGPT-GUI (OpenAI, version GPT-4) with correct partial sentences derived from the a7'ta corpus. At the time of this study, the ChatGPT Interface only supported the GPT-4 model. ChatGPT was then instructed to insert these sentences into the context of a useful sentence comprising five sentences and 200 words. Subsequently, we instructed ChatGPT to replace the correct partial with an incorrect partial. We then used ChatGPT to insert common grammatical errors into the incorrect sentences. For accuracy and relevance, we manually reviewed and validated the generated sentences based on the following requirements: no errors, including correct and incorrect parts in each generated sentence, and consistency with the correct part of the given sentence. Moreover, we manually verified that the correct part has not been replaced with another word that is compatible with the context or one that is synonymous with the correct part. The generated sentences may be inaccurate and may not include all the required information. In this case, we use ChatGPT to generate new sentences that are compatible with the requirements. Although this method was time-consuming, it guaranteed high-quality results. Figures 5–7 show an example of manual ChatGPT sentence generation. Figure 7 illustrates examples of linguistic errors, which are categorized as follows:

- **Misspellings:** An example would be "ربيع الثاني" (The second spring) instead of "ربيع تاني" (Another spring). The omission or replacement of diacritics and miswriting words are common mistakes in typing or pronunciation. This example removes the definite article "ال" and replaces "ث" with "ت".

- **Grammatical errors:** For example, "يقوم المسلمين" (Muslims are doing) rather than (Muslims are doing.) "يقوم المسلمون". A violation of Arabic subject-verb agreement rules is caused by ending "المسلمين" "Muslims" in the incorrect case.

- **Morphological errors:** For instance, "المساجد" instead of "المساجيد". Derivation and incorrect plural forms require better morphological analysis.

- **Punctuation errors:** "قرأن الكريم" "The Holy Quran" without quotation marks or commas. Punctuation errors disrupt sentence flow.

- **Semantic errors:** For example, "أحداث سياسية ليست مرتبطة بالدين" "political events unrelated to religion" are inserted into religious contexts. Context-based meaning generation models are challenged by semantic distortions.

- **Word merging and splitting:** For example, "و إعتدال" instead of "وإعتدال" This example separates the conjunction letter "و" from the word "إعتدال". It should be written without

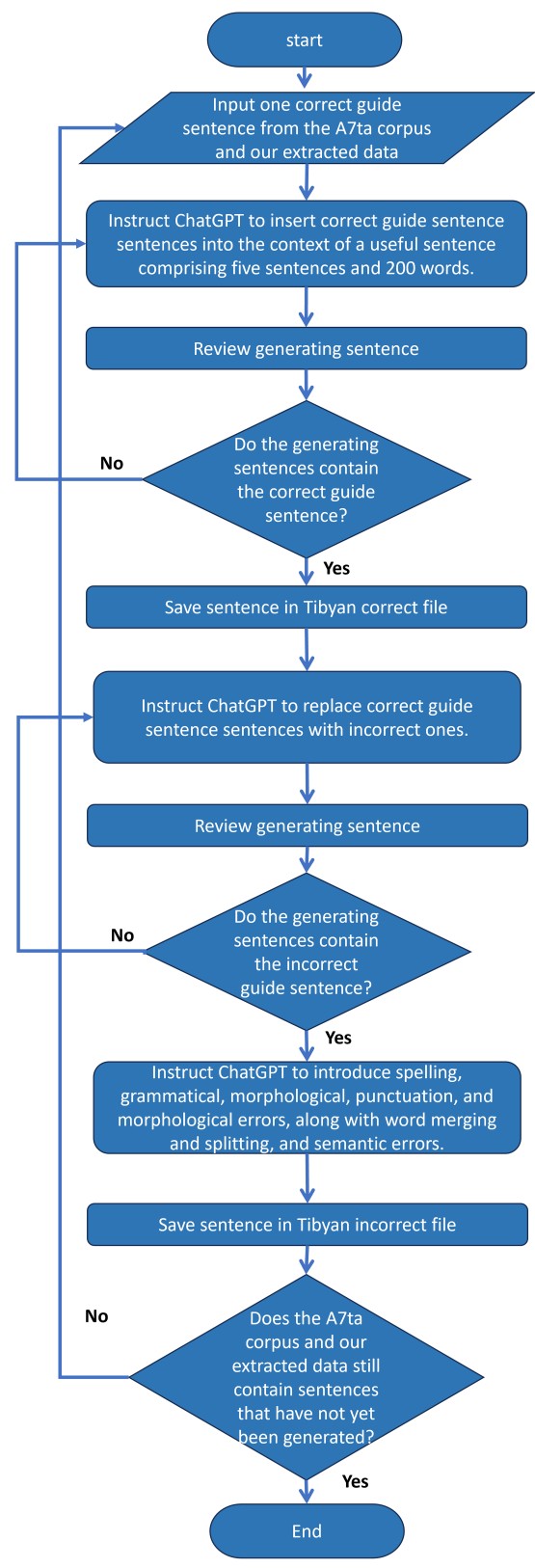

**Figure 4** A flowchart that illustrates the process of generating sentences with ChatGPT.

**Figure 5  An example of manual ChatGPT correct sentence generation.**

**Figure 6  An example of manual ChatGPT incorrect sentence generation.**

any separation, "وإعتدال". These errors are caused by casual mistakes made during writing or typing.

Using this automated approach, we developed a Python script that uses the ChatGPT API (OpenAI, version gpt-3.5-turbo-instruct). To increase the dataset, we launched automated queries on ChatGPT to promote the creation of complete and correct sentences containing the correct guide sentences. Then, we replaced the correct guide sentence with an incorrect one and generated common grammatical errors, resulting in parallel data. For automatic generation, we used all data generated from books and the a7'ta corpus as inputs. It creates five sentences for each correct part of a sentence, which are between 150 and 200 words long. This method is characterized by speed, accuracy, and high-quality data generation.

In the previous sentences, introduce spelling, grammatical, morphological, punctuation, and morphological errors, along with word merging and splitting, and semantic errors.

في الجمل السابقة أنشئ الأخطاء الإملائية والنحوية والصرفية وعلامات الترقيم والمورفولوجية، مع دمج وفصل الكلمات وأخطاء في المعاني؟

1. The month of Rabi' Tani in the Islamic Hijri calendar is a special peroid, where the natur experience flurishing and growh due to the rain and the moderate weathers, but also drought affects some areas.
2. In this spring month, the Muslims revive religious and spiritul activites includeing prayers and reading from Qoran Kareem, and thinking in the verses of Allah, but some forgets the importance of prayrs.
3. The month of Rabi' Thani for Muslims carries religios memory special, where they remamber historical and important events in that time, like political events not related to religion.
4. The month of Rabi Thani is characterise with spirituality in the masajid where the belivers join in religious activity and lessons of religion, but not everyone attends.
5. During the month Rabi Thani, Muslims interacts with the renewed nature and enjoys the beauty of flowersand green trees, while some are busy in other matters not important.

1. يعد شهر ربيع ثاني في التقويم ال هجري الإسلامي فتره مميزا حيث يشهد الطبيعه أزدهار ونمؤ بفضل الامطار و إعتدال الاجواء، لكن ايضا الجفاف يؤثر على بعض المناطق.
2. في هذا الشهر الربيعي يقوم المسلمين بإحياء فعليات دينيه و روحيا تشمل الصلاة والقراءة من قرآن الكريم، والتفكير في أيات اللة ولكن بعض ينسى أهمية الصلاه.
3. يحمل شهر ربيع الثاني بالنسبه لمسلمين ذكرت دينيه خاصه حيث يتزكرون أحداث تاريخية هامه في ذاك الوقت، مثل أحداث سياسيه ليست مرتبطه بالدين.
4. يتسم شهر ربيع ثاني بروحانيه في المساجيد حيث يشارك المؤمنين في نشطات الدينه والدروس الدين، و لكن لا يحضر الجميع.
5. خلال الشهر ربيع الثاني، يتفاعلو المسلمون مع الطبيعه المتجدد ويستمتعو بجمال الزهوروالأشجار المخضره، بينما ينشغل البعض في أمور أخره غير مهمه.

**Figure 7** An example of manual ChatGPT generating grammatical errors in an incorrect sentence.
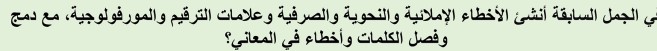

In conclusion, GPT-3.5 adequately performed the automated generation tasks in this study. In addition, preliminary experiments have shown that GPT-3.5's outputs are sufficient for generating high-quality parallel sentences.

## Human annotation

We engaged professional annotators to ensure that the data generated by ChatGPT is correct and error-free. We only provide annotators with data generated by ChatGPT assumed to be correct and part of a corrected sentence (guide sentences) extracted from books, whereas incorrect data are kept without an audit. To ensure the reliability of the human annotation, annotation was performed in two phases.

In the first phase, we instructed the annotator to follow the same guidelines and rules developed by *Zaghouani et al. (2014)* to ensure the uniformity of human annotation and compatibility with previous standards (QALB-14 and QALB-15). We also instructed annotators to correct the morphology, punctuation, spelling, syntax, word choice, and dialectal usage within a given sentence without affecting the wording. Annotators are only required to specify the appropriate corrective action during annotation and not the type of error. Moreover, it instructs annotators to keep the correct parts of sentences extracted from books, without modifying their wording only adapting to the context if necessary. We instructed the annotator to highlight any repeated sentence or word that needs to be removed with a yellow marker instead of deleting it. As soon as we received annotated correct data from the annotator, we manually removed any repeated words and sentences in both the correct and incorrect files to ensure compatibility between parallel data and to make sure all copies were consistent. After the first phase, the following comments were received:

- Some words and phrases are repeated and they should be deleted from the sentences.
- A few words were inserted into sentences that were irrelevant to the context. This should be replaced with a more appropriate word or deleted.
- Some phrases contain religious transgressions or defects from legal, historical, or even realistic standpoints; therefore, they must be changed or deleted. Religious transgressions include acts such as incorrectly attributing statements to the Prophet Muhammad, peace and blessings be upon him. Historical defects such as falsely attributing untrue characteristics to famous and prominent figures like Khalid ibn Al-Walid, such as stinginess, frailty, or similar characteristics.

We then proceed to the second phase of annotation. A qualified linguistic annotator experienced in Islamic history and jurisprudence reviews the text to ensure that sentences are accurate and error-free and modify any questionable sentences. To ensure that a word fits the context, we instructed the annotator to delete or modify the word to fit the context without modifying sentence wording, remove duplicate words if separated by conjunctions, and remove repetitive phrases. Furthermore, if they find a phrase that is not related to context or contains transgressions, they request to change it by a token or phrase related to context and highlight it for updating later in incorrect sentence pairs. This process aimed to achieve high data accuracy and quality through the efforts of professional annotators.

## EXPERIMENT

### Setting

We used gpt-3.5-turbo-instruct for the ChatGPT API. The maximum sequence length was 1,400, the target length was 200, the temperature was 0.8, and the number of sentences generated was five. The prompt consisted of correctly guided sentences from books. The total number of guide sentences was 3,627.

### Implementation details

Implementation was carried out in Python using the OpenAI library and langid for language identification. We first use the following promote "Create a useful sentence consisting of target length words using the guide correct sentence without any changes in the letters or semantic of the phrase: correct_guide" to generate a full corrected sentence. The phrase in our previous promotion was a correct guide sentence, and we instructed the ChatGPT API not to change any letters or semantics so that the phrase's meaning would remain the same, because in Arabic, changing one letter could alter the meaning of the entire sentence. After executing the previous promotion, we obtained five sentences that all included the correct guide sentence separated by a full stop in one line. The generated sentence may not be complete and may end in an incomplete word. To ensure the generation of complete sentences, we identified the maximum sequence length as 1,400; therefore, the generated sentences were not all of the same length but varied from 200 to 1,400. We then applied post-processing to ensure proper formatting and punctuation of the resulting sentences. After that, to construct an incorrect sentence, we only search for

the correct guide in the correct sentence and replace it with the incorrect guide while maintaining the sentence's structure. Finally, we also promote ChatGPT to generate errors in an incorrect sentence pair only by using the following prompt: "In the following sentence, please add spelling, grammar, morphology, punctuation, semantic, and morphology errors, as well as merging and separating words". We used the previous promotion to ensure that the erroneous sentences were diverse and contained a wide range of errors in addition to those derived from books.

### Evaluation metric

For text generation models such as ChatGPT, BERTScore is used to calculate the precision, recall, and F1-scores by comparing the generated text to a target text. BERTScore (*Zhang et al., 2019*) is a metric that measures the similarity between two pieces of text using contextual embeddings from BERT (Bidirectional Encoder Representations from Transformers). Precision measures the number of relevant (correct) words in the generated text. The recall value is the percentage of relevant words in the text generated by those in the reference text. In F1-scoring, precision and recall were balanced, providing a balanced measure.

## RESULTS

The corrected and uncorrected sentences generated by ChatGPT were compared with sentences corrected by a human after annotation. We demonstrated the effectiveness of ChatGPT in generating error-free human-like sentences. Compared to human-corrected sentences, ChatGPT-generated corrected sentences resembled those produced by humans after annotation, as listed in Table 3. The ChatGPT model constructs sentences that follow a natural flow and coherence, similar to human speech. By rigorously analyzing ChatGPT's output, we demonstrated its remarkable ability to produce sentences with accuracy and naturalness comparable to human corrections. A demonstration of ChatGPT's ability to seamlessly integrate linguistic nuances and grammatical rules, ultimately delivering outputs indistinguishable from those produced by humans, serves as an advancement in natural language processing.

## ANALYSIS

In this section, we highlight some of the statistics from our data. We then used the ARETA tool (*Belkebir & Habash, 2021*) to analyze the types of errors when receiving two data files, one with errors and the other without errors, and determine the type of error. ARETA is a system for extracting and annotating Arabic error types. ARETA aims to address the complex and unique challenges of Arabic while being inspired by ERRANT (*Bryant, Felice & Briscoe, 2017*). Natural language processing (NLP) techniques and Arabic morphological analyzers are used to analyze a comprehensive database of grammatical and linguistic errors. The system runs unsupervised, meaning it does not require prior training or ongoing human involvement. Also, it can be applied to a wide range of texts, such as the QALB 2014 competition entries, for evaluating linguistic error correction models.

**Table 3  The BERTScore of correct and incorrect in generating sentences.**

|  | Precision | Recall | F1 |
|---|---|---|---|
| Correct generated sentences | 0.97% | 0.97% | 0.97% |
| Incorrect generated sentences | 0.88% | 0.89% | 0.88% |

**Table 4  General statistics for Tibyan corpus.**

|  | Correct data | Incorrect data |
|---|---|---|
| Lines | 6,191 | 6,191 |
| Words | 618, 598 | 604, 592 |
| Average sentence length | 99.91 | 97.65 |
| Average token length | 4.84 | 4.88 |
| Unique tokens | 71, 976 | 81, 905 |

The ARETA tool is based on *Alfaifi & Atwell (2014)* comprehensive error classification, which classified 29 Arabic language error tags. The ARETA tool includes two modifications to Afifi's comprehensive error classification system. First, merging (MG) and splitting (SP) errors were added to accommodate one-to-many corrections. Furthermore, they removed all other error tags such as OO, MO, XO, SO, and PO, representing orthographic, morphological, syntactic, semantic, and punctuation errors, respectively. Therefore, there were seven classes and 26 error tags in the ARETA taxonomy.

## General statistics and observations

Table 4 summarizes the general statistics of the Tibyan Corpus . The total number of words was 618,598 for correct data and 604,592 for incorrect data. The average sentence length indicated the number of words in each sentence. In the correct data, there are approximately 99.92 words, whereas in incorrect data, there are approximately 97.66 words. The "Average Token Length" shows the average number of characters in each token. In the correct data, there are approximately 4.85 characters per word, whereas in the incorrect data, there are approximately 4.88 characters per word. A "unique token" shows several unique tokens (or words). The error-free data contained 71,976 unique tokens, whereas the error-containing data contained 81,905 unique tokens.

## Analysis of error type before data augmentation

In this section, we analyze the existing error types before data augmentation for both the A7'ta corpus and our extracted data from the books described in the data collection section, using the ARETA tool. The A7'ta corpus consists of 466 sentences and 2,208 tokens. According to the ARETA tool, 22 error types exist in the a7'ta corpus, as listed in Table 5. The error rate is 33%. The corpus lacks four types of errors: lengthening short vowels (OG), shortening long vowels (OS), merged words (MG), and words that are split

**Table 5  Analysis of error type before data augmentation.**

|  | Tag | Error description | A7'ta |  | Our extracted data |  | Both |  |
|---|---|---|---|---|---|---|---|---|
| Orthography | OA | Alif, Ya & Alif-Maqsura | 4 | 0% | 20 | 0% | 24 | 0% |
|  | OC | Char order | 1 | 0% | 12 | 0% | 13 | 0% |
|  | OD | Additional char | 22 | 1% | 176 | 1% | 198 | 1% |
|  | OG | Lengthening short vowels | 0 | 0% | 0 | 0% | 0 | 0% |
|  | OH | Hamza errors | 69 | 4% | 71 | 1% | 140 | 1% |
|  | OM | Missing char(s) | 12 | 1% | 90 | 1% | 102 | 1% |
|  | ON | Nun & Tanwin confusion | 1 | 0% | 12 | 0% | 13 | 0% |
|  | OR | Char replacement | 19 | 1% | 375 | 3% | 394 | 3% |
|  | OS | Shortening long vowels | 0 | 0% | 0 | 0% | 0 | 0% |
|  | OT | Ha/Ta/Ta-Marbuta confusion | 3 | 0% | 17 | 0% | 20 | 0% |
|  | OW | Confusion in Alif Fariqa | 3 | 0% | 0 | 0% | 3 | 0% |
| Morphology | MI | Word inflection | 40 | 2% | 100 | 1% | 140 | 1% |
|  | MT | Verb tense | 2 | 0% | 18 | 0% | 20 | 0% |
| Syntax | XC | Case | 160 | 9% | 872 | 7% | 1,032 | 8% |
|  | XF | Definiteness | 20 | 1% | 32 | 0% | 52 | 0% |
|  | XG | Gender | 20 | 1% | 156 | 1% | 176 | 1% |
|  | XM | Missing word | 33 | 2% | 226 | 2% | 259 | 2% |
|  | XN | Numbers in plural, dual, and singular | 12 | 1% | 51 | 0% | 63 | 0% |
|  | XT | Unnecessary word | 69 | 4% | 684 | 6% | 753 | 6% |
| Semantics | SF | Conjunction error | 9 | 1% | 3 | 0% | 12 | 0% |
|  | SW | Word selection error | 33 | 2% | 681 | 6% | 714 | 5% |
| Punctuation | PC | Punctuation confusion | 6 | 0% | 1 | 0% | 7 | 0% |
|  | PM | Missing punctuation | 16 | 1% | 11 | 0% | 27 | 0% |
|  | PT | Unnecessary punctuation | 3 | 0% | 25 | 0% | 28 | 0% |
| Merge | MG | Words are merged | 0 | 0% | 4 | 0% | 4 | 0% |
| Split | SP | Words are split | 0 | 0% | 0 | 0% | 0 | 0% |
| Unknown | UNK | Unknown errors | 14 | 1% | 194 | 2% | 208 | 2% |
| Comb. | - | Error combinations | 49 | 3% | 400 | 3% | 449 | 3% |
|  |  |  | 571 | 33% | 3,831 | 31% | 4,402 | 32% |

(SP). Owing to an insufficient number of words, a limited number of sentences, and the lack of focus on these types of errors in the a7'ta corpus. Moreover, we observed that lengthening short vowels (OG) error types appeared in combination with Additional Char (OD) error types. Despite its low frequency, it appears only three times. Figure 8 shows the types of combination errors and their frequencies. The ARETA tool generates these errors automatically, and it is common for a single word in Arabic to contain multiple errors.

Our extracted data from the books described in "Data Collection" consists of 3,166 sentences and 12,407 tokens. According to the ARETA tool, 22 error types exist in our data, as listed in Table 5. The error rate is 31%. The corpus lacks four types of error: lengthening short vowels (OG), shortening long vowels (OS), confusion in Alif Fariqa (OW), and words that are split (SP). Moreover, a token may contain more than one type of

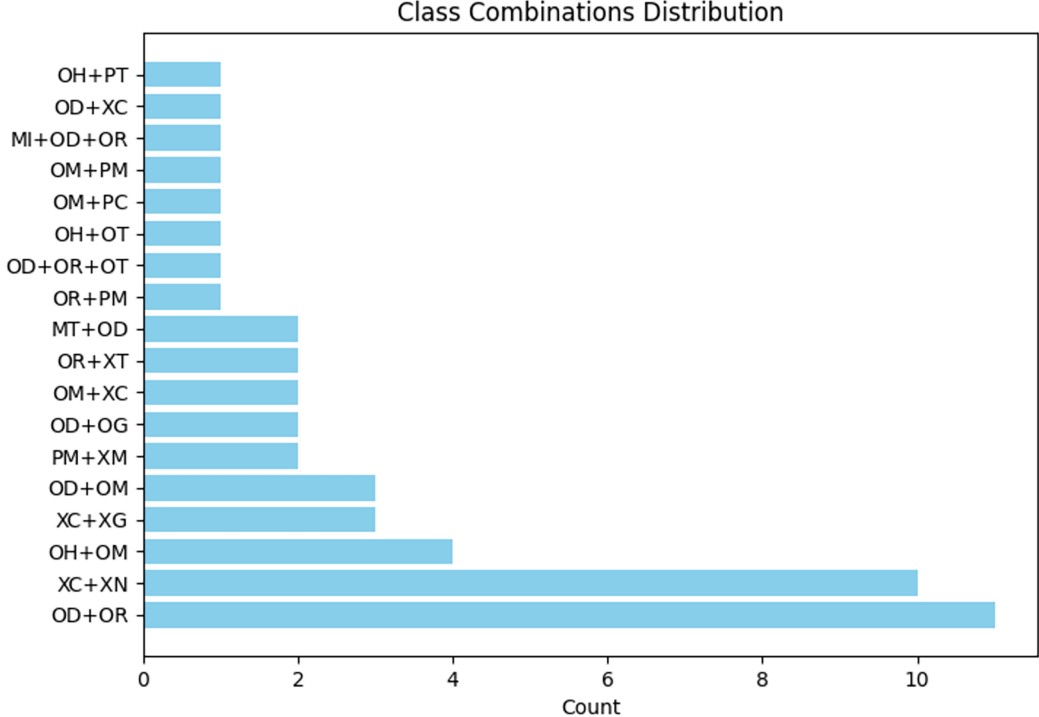

**Figure 8  Types of combination errors and their frequencies in A7'ta corpus.**

error. Figure 9 shows the types of combination errors and their frequencies. We found that OD and OR error types usually existed together at 102 frequencies, followed by OH and OM at 65 frequencies. Another error combination exists at frequencies less than 20. Moreover, we observed that low-frequency error types appeared in combination with other types, such as OD and OG.

When the data were combined, the total error rate was 32%. There were 23 types of errors and a lack of three types: lengthening short vowels (OG), shortening long vowels (OS), and words that are split (SP). This is due to the lack of a sufficient number of words and a limited number of sentences. As the tool did not accurately classify some semantic errors, 449 unknown errors were found, such as when we replaced an MSA token with a dialect or foreign word, using a word that differed from its meaning, or when more than one error was present in a phrase. Figure 10 shows the top ten types of combination errors and their frequencies. We observed that some error types usually exist together, such as OD with OR and OH and OM error types.

## Analysis of error type after data augmentation

In this section, we analyze the error type after data augmentation, and before and after human annotation.

### Analyze the error type before human annotation

The error rate increased by 7% for the a7'ta corpus (manual generation), 13% for our extracted data, and 11% for all the data, as listed in Table 6. Our corpus contained 23 types

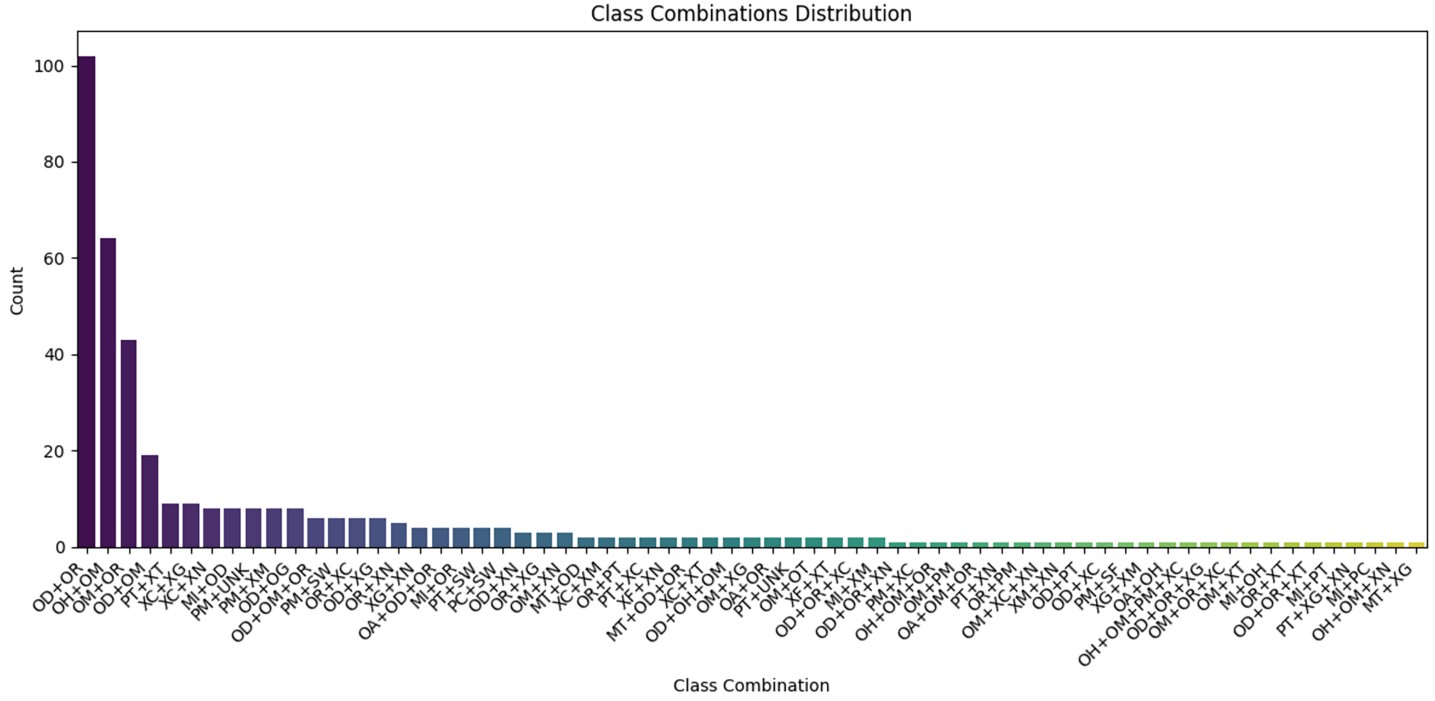

**Figure 9 Types of combination errors and their frequencies in our extracted data.**

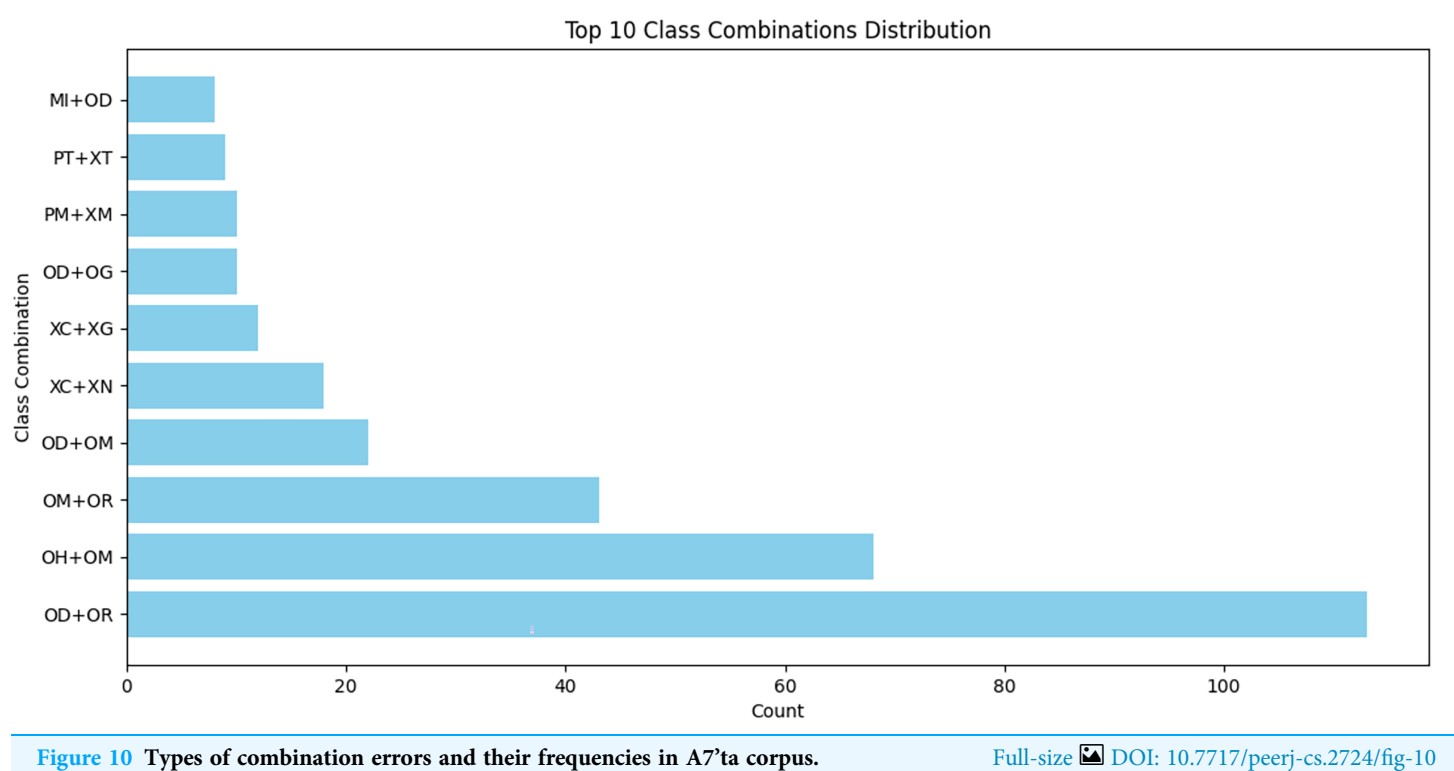

**Figure 10 Types of combination errors and their frequencies in A7'ta corpus.**

**Table 6 Analysis of error type after data augmentation before human annotation.**

| | Tag | Error description | A7'ta corpus (Manual) | | Our data + A7'ta (Automatic) | | Tibyan corpus | |
|---|---|---|---|---|---|---|---|---|
| Orthography | OA | Alif, Ya & Alif-Maqsura | 1,615 | 3% | 20,170 | 4% | 21,785 | 4% |
| | OC | Char order | 447 | 1% | 197 | 0% | 644 | 0% |
| | OD | Additional char | 289 | 1% | 354 | 0% | 643 | 0% |
| | OG | Lengthening short vowels | 0 | 0% | 0 | 0% | 0 | 0% |
| | OH | Hamza errors | 3,881 | 8% | 45,210 | 9% | 49,091 | 9% |
| | OM | Missing char(s) | 157 | 0% | 4,333 | 1% | 4,490 | 1% |
| | ON | Nun & Tanwin Confusion | 0 | 0% | 4 | 0% | 4 | 0% |
| | OR | Char Replacement | 2,446 | 5% | 1,894 | 0% | 4,340 | 1% |
| | OS | Shortening long vowels | 0 | 0% | 0 | 0% | 0 | 0% |
| | OT | Ha/Ta/Ta-Marbuta Confusion | 8,608 | 17% | 56,273 | 12% | 64,881 | 12% |
| | OW | Confusion in Alif Fariqa | 10 | 0% | 135 | 0% | 145 | 0% |
| Morphology | MI | Word inflection | 122 | 0% | 328 | 0% | 450 | 0% |
| | MT | Verb tense | 1 | 0% | 38 | 0% | 39 | 0% |
| Syntax | XC | Case | 284 | 1% | 8,081 | 2% | 8,365 | 2% |
| | XF | Definiteness | 314 | 1% | 1,193 | 0% | 1,507 | 0% |
| | XG | Gender | 18 | 1% | 179 | 0% | 197 | 0% |
| | XM | Missing word | 18 | 0% | 256 | 0% | 274 | 0% |
| | XN | Numbers in plural, dual, and singular | 26 | 0% | 124 | 0% | 150 | 0% |
| | XT | Unnecessary word | 231 | 0% | 602 | 0% | 833 | 0% |
| Semantics | SF | Conjunction error | 4 | 0% | 71 | 0% | 75 | 0% |
| | SW | Word selection error | 292 | 1% | 2,951 | 1% | 3,243 | 1% |
| Punctuation | PC | Punctuation confusion | 269 | 1% | 31,822 | 7% | 32,091 | 6% |
| | PM | Missing punctuation | 307 | 1% | 6 | 0% | 313 | 0% |
| | PT | Unnecessary punctuation | 12 | 1% | 508 | 0% | 520 | 0% |
| Merge | MG | Words are merged | 78 | 0% | 2,112 | 0% | 2,190 | 0% |
| Split | SP | Words are split | 0 | 0% | 0 | 0% | 0 | 0% |
| Unknown | UNK | Unknown errors | 0 | 0% | 0 | 0% | 0 | 0% |
| Comb. | - | Error combinations | 1,635 | 3% | 36,940 | 8% | 38,575 | 7% |
| | | | 20,559 | 40% | 213,784 | 44% | 235,055 | 43% |

of errors. Three types of errors do not exist in the Tibyan corpus: OG, OS, and SP. Only four instances of the ON error type were in the Tibyan corpus. Although some errors exist at high frequencies such as OT, OH, PC, and OA, others exist at low frequencies such as ON, MT, and SF. Figure 11 shows the error combinations of the top five low-frequency classes. The types of errors that appear in small percentages appear in combination with other types in varying percentages, as shown in Fig. 10. In conclusion, all types of errors appeared in varying proportions, either alone or in combination. Figure 11 shows the error combination for the top five low-frequency classes. The types of errors that appeared in small percentages appear combined with other types in varying percentages, as shown in Fig. 10. In conclusion, all types of errors appeared in varying proportions, either alone or in combination.

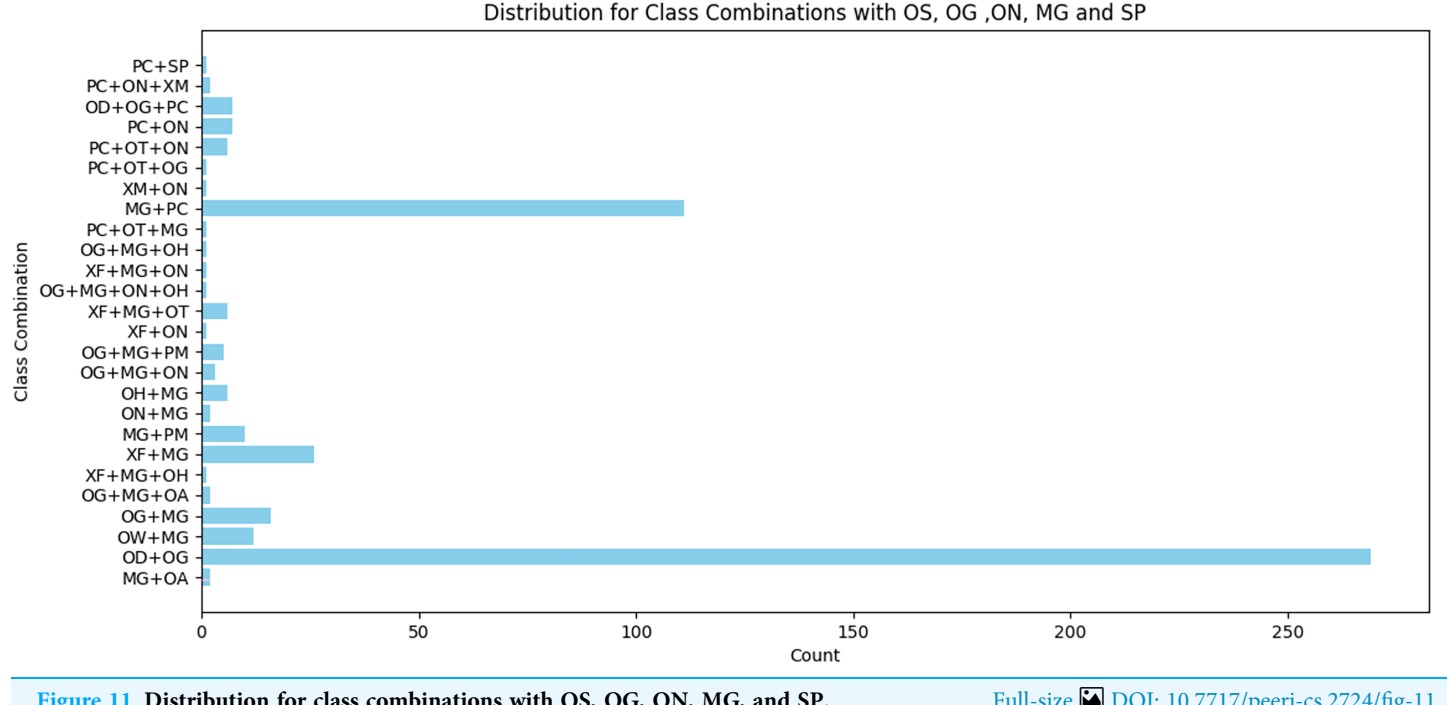

**Figure 11 Distribution for class combinations with OS, OG, ON, MG, and SP.**

### Analyze the error type after human annotation

In comparison with analyzing the error type following data augmentation before human annotation, we notice that a7'ta corpus (manual generation) has an error rate of 8%, our extracted data has an error rate of 5%, and all our data has an error rate of 6%, as listed in Table 7. Our corpus contained 26 types of errors. Furthermore, it contains two types of errors: OG and OS. The ON error types in the corpus increased to 527 instances. Although some errors exist at high frequencies such as OT, OH, PC, and OA, others exist at low frequencies such as ON, MT, and SF. Figure 12 shows the error combinations of the top five low-frequency classes. The types of errors that appeared in small percentages were combined with other types of errors in varying percentages. In conclusion, all error types appeared in varying proportions, either alone or in combination.

A strength of the Tibyan Corpus is that it includes common errors by native Arabic speakers. There are several types of errors, including Alif, Ya, and Alif-Maqsura (OA), Hamza errors (OH), Ha/Ta/Ta-Marbuta confusion (OT), and punctuation confusion (PC). Despite the complexities of Arabic orthography and grammar, these errors are challenging for native speakers.

Native speakers, however, are more likely to make errors, such as char order (OC), additional char (OD), missing char (OM), char replacement (OR), nun and tanwin confusion (ON), confusion in Alif Fariqa (OW), merge (MG), and split (PS). It is more common for these errors to arise from typing than from a lack of language understanding. Because these errors are usually straightforward and follow predictable patterns, modern applications and spell checkers are generally effective at identifying and resolving them.

**Table 7 Analyze the error type after human annotation.**

| | Tag | Error description | A7'ta Corpus (Manual) | | Our data + A7'ta (Automatic) | | Tibyan Corpus | |
|---|---|---|---|---|---|---|---|---|
| Orthography | OA | Alif, Ya & Alif-Maqsura | 1,428 | 3% | 19,543 | 4% | 20,971 | 4% |
| | OC | Char Order | 432 | 0% | 181 | 0% | 613 | 0% |
| | OD | Additional Char | 236 | 1% | 605 | 0% | 841 | 0% |
| | OG | Lengthening short vowels | 94 | 0% | 426 | 0% | 520 | 0% |
| | OH | Hamza errors | 3,623 | 7% | 44,387 | 10% | 48,010 | 9% |
| | OM | Missing char(s) | 129 | 0% | 4,463 | 1% | 4,592 | 1% |
| | ON | Nun & Tanwin Confusion | 10 | 0% | 517 | 0% | 527 | 0% |
| | OR | Char Replacement | 2,318 | 5% | 2,298 | 1% | 4,616 | 1% |
| | OS | Shortening long vowels | 105 | 0% | 55 | 0% | 160 | 0% |
| | OT | Ha/Ta/Ta-Marbuta Confusion | 8,069 | 16% | 53,119 | 12% | 61,188 | 12% |
| | OW | Confusion in Alif Fariqa | 9 | 0% | 112 | 0% | 121 | 0% |
| Morphology | MI | Word inflection | 276 | 1% | 1,318 | 0% | 1,594 | 0% |
| | MT | Verb tense | 14 | 0% | 95 | 0% | 109 | 0% |
| Syntax | XC | Case | 639 | 1% | 11,372 | 2% | 12,011 | 2% |
| | XF | Definiteness | 378 | 1% | 1,615 | 0% | 1,993 | 0% |
| | XG | Gender | 81 | 0% | 709 | 0% | 790 | 0% |
| | XM | Missing word | 185 | 0% | 1,713 | 3% | 1,898 | 0% |
| | XN | Numbers in plural, dual, and singular | 61 | 0% | 391 | 0% | 452 | 0% |
| | XT | Unnecessary word | 1,011 | 2% | 3,666 | 1% | 4,677 | 1% |
| Semantic | SF | Conjunction error | 94 | 0% | 112 | 0% | 206 | 0% |
| | SW | Word selection error | 1,527 | 3% | 4,430 | 1% | 5,957 | 1% |
| Punctuation | PC | Punctuation confusion | 497 | 1% | 18,731 | 4% | 19,228 | 4% |
| | PM | Missing punctuation | 435 | 1% | 3,641 | 1% | 4,076 | 1% |
| | PT | Unnecessary punctuation | 16 | 0% | 1,251 | 0% | 1,267 | 0% |
| Merge | MG | Words are merged | 77 | 0% | 0 | 0% | 1,420 | 0% |
| Split | SP | Words are split | 5 | 0% | 102 | 0% | 107 | 0% |
| Unknown | UNK | Unknown Errors | 0 | 0% | 0 | 0% | 0 | 0% |
| Comb. | - | Error Combinations | 2,663 | 5% | 54,998 | 12% | 57,413 | 11% |
| | | | 23,305 | 48% | 224,737 | 49% | 308,408 | 49% |

Furthermore, some errors appear less frequently because they are uncommon among native speakers and are more likely to be made by second-language learners. For example, lengthening short vowels (OG), shortening long vowels (OS), and gender (XG). Native language writers usually use correct words and avoid grammatical mistakes. Language processing tools can be developed more efficiently by distinguishing between errors typical of native speakers and those typical of second-language learners.

## APPLICATIONS AND IMPLICATIONS

In the technology and artificial intelligence world, the Tibyan corpus represents an advanced step toward enhancing Arabic language processing capabilities. Tibyan corpus holds great promise in a variety of fields that could be instrumental in improving our daily lives and preparing us for the future.

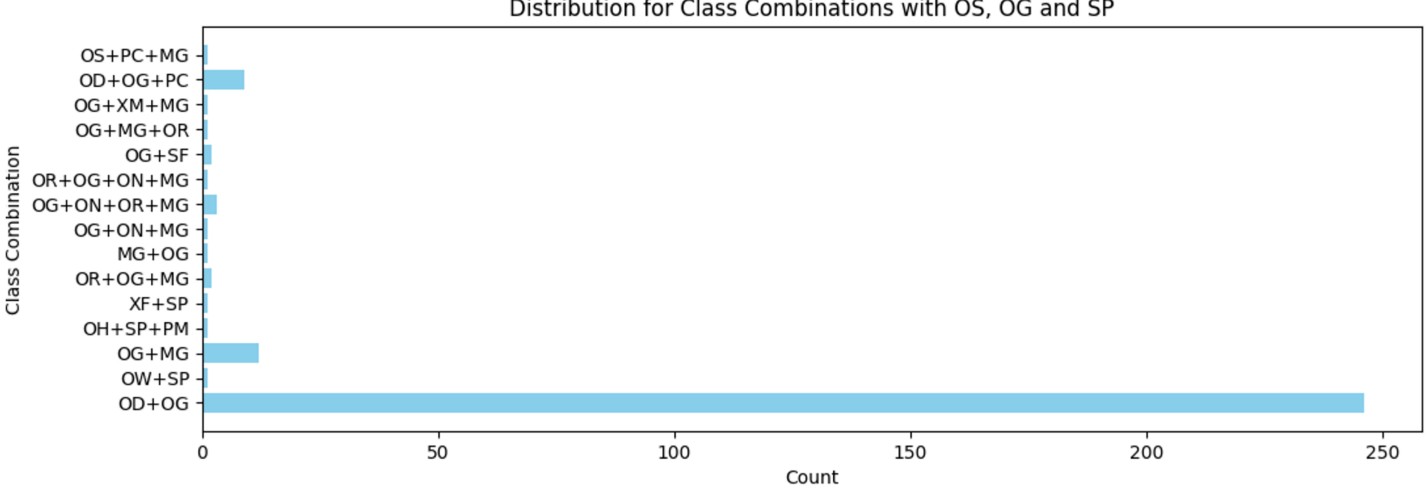

**Figure 12** Distribution for class combinations with OS, OG, and SP.

**Self-Learning and Education**. This corpus can be used by Arabic language students to develop intelligent educational systems that provide automatic feedback and corrections. Learners could acquire writing skills accurately and seamlessly with an educational platform that understands and corrects grammatical errors in an interactive and personalized way.

**Development of advanced artificial intelligence systems**. It facilitates the development of advanced language models that can understand human errors and produce accurate and high-quality texts. A variety of artificial intelligence systems can be trained to work in areas such as text proofreading, creative writing, and writing assistance in the workplace.

**Enhancing natural language processing research**. The Arabic corpus provides a strong benchmark for comparing different models' performance in correcting grammatical errors. Natural language processing research can benefit from it since it can help researchers develop more efficient and better models for analyzing Arabic text.

**Offers content and media services**. The corpus can be used in text editing applications and improve the quality of media content in the digital age. With this corpus, grammar correction systems can improve text accuracy and reduce errors in any type of text, from blogs to academic articles.

**The promotion of Arabic's universal use**. As a result of this work, the Arabic language can be better understood and corrected by technology, enhancing its global status in fields such as education, business, and technological development.

It enables the development of the Arabic language in the digital age by bridging current challenges and future opportunities.

## LIMITATION

ARETA is the only tool that classifies Arabic grammatical errors accurately at the time of this research. However, the ARETA tool is not always accurate in determining Arabic

errors. ARETA tool sometimes fails to classify words with more than two errors and mark them with X or UNKs, even though the tool already recognizes the various errors present in that word. For example, if a punctuation mark separates two words, and if the first and second words contain errors and there is no space between the punctuation mark and words, it is classified as UNK or X. In addition, we observed that it did not correctly classify words that have split (SP) errors. If the first letter of a word is separated from the rest, it is treated as one word, classified as an unnecessary word "XT" and another word is classified as a missing letter "OM", and they will not be considered one word. Furthermore, merge errors containing n more than one error type were incorrectly classified. To ensure the correct classification of error types, we manually classified the X and UNK errors.

## CONCLUSION

In conclusion, this study aimed to create an Arabic corpus called Tibyan for Grammatical Error Correction. We use a diverse range of Arabic text extracted from books and the a7'ta corpus containing common grammatical errors. The ChatGPT model is then used to generate parallel sentences containing quoted sentences extracted from Arabic books, one with grammatical errors and the other with correct sentences. By engaging linguistic experts and iteratively refining the corpus based on their feedback, we ensured that it represented the real world and was reliable. Ultimately, the Tibyan corpus will enable the development of an accurate and powerful grammatical error correction tool tailored specifically for the Arabic language. Two key aims are addressed by the corpus : error-type coverage and unbalanced error-type classification. The Tibyan corpus contains all types of errors in Arabic, as the sentences were extracted from books, representing a diverse and rich source of errors. Moreover, it achieves a balance between types of errors. All the types of errors appeared in proportion to each other. We will use our corpus to construct a robust Arabic grammatical error correction model in the future. In addition, the number of sentences and tokens can be increased using modern techniques.

## ACKNOWLEDGEMENTS

The researchers greatly appreciate ChatGPT models (versions GPT-4 and GPT-3.5-turbo-instruct) provided by OpenAI that were used to augment the data for grammatical error correction tasks. With the help of these tools, we were able to increase the number of parallel Arabic data and simulate linguistic errors, enhancing the accuracy and diversity of the experimental data.

### Funding

The authors received no funding for this work.

### Competing Interests

The authors declare that they have no competing interests.

## Author Contributions

- Ahlam Alrehili conceived and designed the experiments, performed the experiments, analyzed the data, performed the computation work, prepared figures and/or tables, authored or reviewed drafts of the article, and approved the final draft.
- Areej Alhothali analyzed the data, authored or reviewed drafts of the article, and approved the final draft.

## Data Availability

The data is available at Zenodo: Alrehili, A. (2025). Tibyan-corpus [Data set]. Zenodo. https://doi.org/10.5281/zenodo.14623621.

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
