# Peer review of "Tibyan corpus: balanced and comprehensive error coverage corpus using ChatGPT for Arabic grammatical error correction"

_PeerJ Computer Science, doi:10.7717/peerj-cs.2724_

## Round 0.1 · original submission · Major Revisions

Both reviewers have listed a set of areas for improvement that should be addressed in a round of major revisions before it can be considered for publication.

Reviewer 1 ·

Basic reporting

-The abstract needs to be more concise. Consider removing some of the background information.

-The in-text citations are not formatted accurately; they are difficult to read.

The authors used ChatGPT for augmentation. However, ChatGPT is just an interface, not a model. Open AI GPT 4, GPT-4o, and other models can be accessed using ChatGPT. To avoid ambiguity, the authors should consistently refer to the specific openAI model used.

-Translate prompts and examples from Arabic to English whenever possible (such as in Figure 4).

-Language and formatting require substantial improvements.

Experimental design

-The annotation process is unclear and contains several limitations. The authors should provide the exact instructions given to the annotators. Some of the steps are questionable. For instance, when removing terms like "religious transgressions" they are vague/subjective. There is a lack of objective measures such as inter-annotator agreement to evaluate the reliability of annotations systematically.

-The implications of this study are not discussed. How can the corpus be used in the real world? Is it effective in applications such as education or building information systems?

-The use of GPT 3.5 begs the question as to why the authors decided not to rely on state-of-the-art models like GPT-4 or GPT-4o.

Validity of the findings

no comment

Reviewer 2 ·

Basic reporting

The authors address an interesting topic in Natural Language processing (NLP). The authors conduct a study that aims to develop an Arabic corpus which they call it "Tibyan" for grammatical error correction using ChatGPT. They employed ChatGPT to generate parallel sentences that contains quoted sentences extracted from Arabic books, one with grammatical errors and the other with correct sentences. In general, the paper is well-organized and the authors discuss their proposed approach by presenting most of the necessity background to understand the paper. However, there is some space to improve the manuscript quality. The following comments should be addressed before accepting the paper for publication:
1- There is no need to repeat the definition of the acronyms in the manuscript several times such as LLM and GEC. They are defined several times in the document. It would be betted to add a table that defines all the acronyms in the manuscript.
2- It would be better to explain more about ARETA tool.
3- The sections are not numbered in the manuscript, Therefore, why do you use section number in some places in the manuscript. For example in line 372, you refer to section 3.1 (….our extracted data from the books described in Section 3.1…).
4- It would be better to use flowchart diagram to represent the methodology that was considered in the study and which was discussed in lines (251 – 280).
5- It would be better to write the sentence in line 272 ( Figure 4, Figure 5, and Figure 6 show an example of manual ChatGPT sentence generation.) as (Figures 4, 5, and 6 show an example of manual ChatGPT sentence generation.)
6- Figures 4, 5, and 6 need more explanation and discussion. What about the errors in the meaning of the sentence? For example, sentence 1 in Figures 1. Was the comment/rule in lines 299-300 considered in this case?
7- “Zaghouanietal.(2014)” is repeated two times in line 292.
8- For the results presented in Figures 7, 8 and 10, what was/were the rule(s) for combing the errors? Is this process of combination realistic in Arabic language? Please, explain?
9- Figure 9 is missing in the manuscript.
10- What does the error (XN =number) mean?
11- In the LIMITATION section, the authors discussed the limitation of the ARETA tool, is there any method or another tool that can be used and give more efficient error analysis in Arabic language?

Experimental design

no comment

Validity of the findings

no comment

Additional comments

no comment

---

## Round 0.2 · Minor Revisions

Please address the minor revisions suggested by the reviewers. You may ignore R1's comments about the format as this is due to the template.

Reviewer 1 ·

Basic reporting

The manuscript has substantially improved over the last revision. The authors do notable work in improving the abstract and clarifying aspects of the methodology.

The authors mention contradictory narratives regarding the GPT model selection.

"The generation tasks involved in this study were adequately performed by GPT-3.5, despite
324 GPT-49s advanced capabilities and computational cost considerations. In addition, preliminary
325 experiments have shown that GPT-3.59s outputs are suûcient for generating high-quality parallel
326 sentences."

"In the manual approach, we manually provided ChatGPT-GUI (OpenAI, version
282 GPT-4) with correct partial sentences derived from the a79ta corpus. At the time of this study,
283 the ChatGPT Interface only supported the GPT-4 model."

It is important to be consistent and unambiguous. If the manual approach was GPT-4 and generation was GPT -3.5, it should be clear. Referring to both as ChatGPT creates confusion so I recommend being specific and consistent about the model name everywhere it's referenced.

The authors are encouraged to adjust minor formatting issues. It is customary to use a three-line table (unless the journal has a specific format).

The sections and subsections should be numbered for ease of reading and navigation.

Experimental design

N/A

Validity of the findings

N/A

Reviewer 2 ·

Basic reporting

The authors have addressed most of my comments. However, some minor typos have been noticed as a result of this modification.
1- The paragraph in lines 87-93, has some types:
- Next, it describes  should be Next, we describe
- Then describes our experimental setup  should be Then we describe our experiment setup
- Subsequently, analyzes the type  should be Subsequently, we analysis the type

The same for the remain sentences in the paragraph.

2- The caption of Figure 4 should be:
Flowchart that illustrates the process of generating sentences with ChatGPT

Experimental design

Not applicable

Validity of the findings

Not applicable

Additional comments

Not applicable

---

## Round 0.3 · accepted · Accept

I appreciate the authors for addressing the minor issues raised in the last round of reviews. I have checked these revisions and they are all satisfactory, and therefore I recommend acceptance of this work in its current form.